# Rendering Wood Veneers Flexible and Electrically Conductive through Delignification and Electroless Ni Plating

**DOI:** 10.3390/ma12193198

**Published:** 2019-09-29

**Authors:** Minfeng Chen, Weijun Zhou, Jizhang Chen, Junling Xu

**Affiliations:** 1College of Materials Science and Engineering, Nanjing Forestry University, Nanjing 210037, China; chenmf@njfu.edu.cn (M.C.); zhouwj@njfu.edu.cn (W.Z.); 2Department of Electronic Engineering, The Chinese University of Hong Kong, NT, Hong Kong 999077, China; junlingxu@outlook.com

**Keywords:** wood veneers, flexibility, electrical conductivity, delignification, electroless plating

## Abstract

Wood has unique advantages. However, the rigid structure and intrinsic insulating nature of wood limit its applications. Herein, a two-step process is developed to render wood veneers conductive and flexible. In the first step, most of the lignin and hemicellulose in the wood veneer are removed by hydrothermal treatment. In the second step, electroless Ni plating and subsequent pressing are carried out. The obtained Ni-plated veneer is flexible and bendable, and has a high tensile strength of 21.9 and 4.4 MPa along and across the channel direction, respectively, the former of which is considerably higher than that of carbon cloth and graphene foam. Moreover, this product exhibits high electrical conductivity around 1.1 × 10^3^ S m^−1^, which is comparable to that of carbon cloth and graphene foam, and significantly outperforms previously reported wood-based conductors. This work reveals an effective strategy to transform cheap and renewable wood into a high value-added product that rivals expensive carbon cloth and graphene foam. The obtained product is particularly promising as a current collector for flexible and wearable electrochemical energy storage devices such as supercapacitors and Li-ion batteries.

## 1. Introduction

Recently, the research on wood-based functional materials is emerging as a cutting-edge field [1,2]. As a raw material, wood is abundant, cheap, renewable, biodegradable, and possesses a porous structure. Consequently, the utilization of wood is a key to realizing sustainable development. Strategies for constructing wood-based functional materials can be classified into two categories, i.e., bottom-up assembly and top-down functionality. The bottom-up strategy mainly refers to the controllable assembly of cellulose microfibrils, nanocrystals, and nanofibrils into structures of fiber [3], membrane [4], hydrogel [5,6,7], sponge/aerogel [8], etc., while the top-down strategy makes use of the unique structure of wood that encompasses nanoscale, microscale, and macroscale features [9]. The latter strategy is deemed more efficient and cost-effective [9], and the obtained materials have shown great promise in the applications of electrochemical energy storage (EES) [10], transparent films [11], sensors [12], wave adsorption [13], solar steam generation [14], and so on.

With the development of flexible and wearable electronics, the demands for flexible and electrically conductive current collectors have grown rapidly in recent years. Although the carbonization of wood at a high temperature (e.g., 1000 °C) can generate electrically conductive carbon monolith [15], this methodology requires high energy input and sacrifices the biodegradability of wood, and the obtained carbon monolith has moderate electrical conductivity and poor flexibility. Several other reports have deposited conducting polymers (e.g., polyaniline (PANi) and polypyrrole (PPy)) into the lumen space of wood, whereas the electrical conductivity of the modified wood veneers is not high enough [16,17,18]. In the present study, using natural balsa wood as a raw material, we successfully constructed conductive and flexible wood veneer through hydrothermal treatment, electroless plating, and subsequent high-pressure pressing. The electrical conductivity of this functional wood veneer is comparable to that of carbon cloth and graphene foam, which are widely used as flexible current collectors for EES devices, and significantly higher than that of previously reported wood-based conductors. Our product also has a high tensile strength of 21.9 and 4.4 MPa in two different directions, superior to carbon cloth and graphene foam. It should be noted that carbon foam is rather expensive, while the synthesis of graphene foam is very expensive with a limited product yield. This study proposes a scalable and cost-effective methodology to construct a functional wood veneer that compares favorably with carbon cloth and graphene foam. Besides the advantages mentioned above, the wood fibers in our product can function as an electrolyte reservoir that allows for efficient ion transport [19], making our product ideal substrate for flexible EES devices.

## 2. Materials and Methods

### 2.1. Preparation of the Ni-Plated H-Veneer

First, natural balsa wood was cut into veneers with a size of around 30 × 20 × 1.1 mm. The veneer was immersed in an aqueous solution containing 2.5 M NaOH and 0.4 M Na_2_SO_3_, transferred to a Teflon-lined autoclave, and then maintained at 130 °C for 7 h. After hydrothermal treatment, the veneer was washed with boiling deionized water (DI water) three times, and then freeze-dried. The obtained veneer is called H-veneer for short. The H-veneer without freeze-drying was immersed in an aqueous solution containing 0.5 M HCl, 0.05 M SnCl_2_, and several Sn granules for 10 min, followed by rinsing with DI water three times. The veneer was then immersed in an aqueous solution containing 0.036 M HCl and 20 μg/mL PdCl_2_ for 10 min, followed by rinsing with DI water three times. Subsequently, the veneer was immersed in an aqueous solution containing 0.034 M sodium citrate, 0.561 M NH_4_Cl, 0.0712 M NiSO_4_, and 0.283 M NaH_2_PO_2_ (the pH of the solution was adjusted to 9‒10 using concentrated ammonia water) for 6 h, followed by washing with DI water three times. After freeze-drying, the veneer was pressed at 10 MPa, producing a Ni-plated H-veneer.

### 2.2. Material Characterization

The morphologies of the veneers were observed on a JEOL JSM-7600F field emission scanning electron microscope (FE-SEM) (JEOL, Tokyo, Japan) equipped with an energy dispersive X-ray spectroscopy (EDS) detector. The crystallographic information, phase purity, and chemical composition of the veneers were recorded by Rigaku Ultima IV powder X-ray diffractometer (XRD) (Rigaku, Tokyo, Japan) with Cu Kα radiation source *(λ* = 0.1540598 nm), Thermo Scientific DXR5 32 Raman spectrometer (*λ* = 532 nm) (Thermo Scientific, Miami, OK, USA), and Thermo Scientific ESCALAB 250Xi X-ray photoelectron spectrometer (XPS) (Thermo Scientific, Miami, OK, USA). Mechanical tests were carried out on a SHIMADZU AG-Xplus universal testing machine (SHIMADZU, Tokyo, Japan) at a strain rate of 1 mm min^−1^. The electrical conductivity values were collected by a CHI 660E electrochemical workstation (Chenhua, China) and ST2253 four-point probe tester (Suzhou Jingge, China) at room temperature.

## 3. Results and Discussion

The procedure of endowing wood veneers with good flexibility and high electrical conductivity is schematically illustrated in Figure 1a. Natural balsa wood veneers with a size of around 30 × 20 × 1.1 mm were used as the starting material. The veneer was hydrothermally treated in a mixed aqueous solution containing NaOH and Na_2_SO_3_. As is known, the lumen walls of wood consist of cellulose, hemicellulose, and lignin. During the hydrothermal treatment, most of the lignin and hemicellulose were removed, while most of the cellulose was preserved, largely owing to their different stabilities in the sulfite-containing alkaline solution [1,20]. The removal of lignin and hemicellulose can be verified by the 52.7% weight loss of the pristine veneer (denoted P-veneer) after hydrothermal treatment. The hydrothermally treated veneer is denoted as the H-veneer. Then, Ni was deposited onto the lumen walls of the H-veneer through electroless plating, followed by pressing at 10 MPa. The top of Figure 1b shows photographic images of the P-veneer, H-veneer without pressing, H-veneer after pressing, and Ni-plated H-veneer (from left to right). The black color of the Ni-plated H-veneer confirms the successful Ni plating. Compared to the P-veneer, the treated veneers have similar sizes on the surface (marked in Figure 1a). In contrast, the cross-sectional thickness was greatly reduced for the H-veneer after pressing and the Ni-plated H-veneer, being ca. 26% and 27% that of the P-veneer, respectively. The thickness reduction is associated with the delignification of the veneer, which is beneficial to the mechanical strength [1]. As is shown in the bottom of Figure 1b, both the H-veneer after pressing and the Ni-plated H-veneer are highly flexible in the direction perpendicular to the surface. On the contrary, the P-veneer is easily broken under bending. The surface SEM images of the four veneer samples are compared in Figure 1c–g. In the P-veneer (see Figure 1c), some substances are observed to cover the aligned fibrils. After hydrothermal treatment, most of these substances are removed in the H-veneer (Figure 1d), and the fibril edges become more distinct. After pressing, the fibrils are flattened and densely packed together (Figure 1e). After electroless plating and subsequent pressing, the alignment of the fibrils is maintained (Figure 1f), while Ni particles are observed to grow on the surface of these fibrils (Figure 1g). The plating of Ni can also be confirmed by the EDS spectrum in the inset of Figure 1e. The cross-sectional SEM images of the four samples are shown in Figure 1h–p. There are numerous lumens (tubular channels) along the wood growth direction in the P-veneer (Figure 1h,i). By the removal of lignin and hemicellulose, the lumen walls seem to become softer (Figure 1j,k). After pressing, the lumens collapse and the walls are tightly piled in the H-veneer (Figure 1l,m). As for the Ni-plated H-veneer (Figure 1n–p), its morphology is similar to that of the H-veneer after pressing, except that the Ni particles are uniformly deposited on the lumen walls of the Ni-plated H-veneer.

The samples are further characterized by XRD analysis, as shown in Figure 2a. The P-veneer and H-veneer reveal peaks at 14.8°, 16.2°, 22.4°, and 34.2°, corresponding to the (1−10), (110), (200), and (004) crystalline planes of cellulose I, respectively [21,22]. Hemicellulose and lignin cannot be detected by XRD due to their amorphous nature. After electroless plating, a new peak at 44.5° appears, which is assigned to the (011) peak of metallic Ni (JCPDS 45-1027). The Raman spectra of the samples are presented in Figure 2b. The H-veneer exhibits three peaks that can be well indexed as cellulose, together with one large hump belonging to lignin [23,24]. In the P-veneer, the peaks arising from cellulose are located at slightly different positions from that of the H-veneer, which might result from the existence of hemicellulose in the P-veneer. In addition, it can be seen that the peak intensity ratio of lignin/cellulose in the H-veneer is much lower than that in the P-veneer, implying that lignin is substantially removed during hydrothermal treatment. As for the Ni-plated H-veneer, the abovementioned peaks almost disappear, owing to the fact that the lumen walls are covered by metallic Ni.

XPS measurements were conducted to investigate the surface compositions of the veneer samples. As shown in Figure 3a, the P-veneer is mainly comprised of C and O elements (H is undetectable by XPS), and contains trace amounts of Si, S, Ca, and N. After hydrothermal treatment, only C and O can be detected in the H-veneer. After electroless plating, strong signals from Ni are observed in the Ni-plated H-veneer, indicating the successful deposition of Ni. Figure 3b gives high-resolution C 1s XPS spectra, which are deconvoluted into four sub-peaks that are attributed to C‒C (and C‒H), C‒O‒C (and C‒OH), O‒C‒O (and C=O), and O‒C=O bonds, respectively [25]. As cellulose possesses more C‒O‒C and C‒OH bonds compared to lignin, the subpeak corresponding to these bonds in the H-veneer delivers a much higher intensity than that in the P-veneer. Moreover, the peak intensity corresponding to O‒C=O in the H-veneer is negligible, suggesting efficient delignification. As for the Ni-plated H-veneer, its C 1s spectrum differs from that of the H-veneer, probably due to the formation of a Ni layer on its surface, since XPS characterization is highly sensitive to the material surface.

Figure 4a shows tensile strain‒stress curves of four veneer samples along the channel direction. For comparison, carbon cloth (W0S1002, CeTech Co., Ltd., Taiwan, China) was tested under the same condition. This same carbon cloth has been reported as the current collector for EES in many other reports [26,27]. The P-veneer has a tensile strength of 3.5 MPa, which decreases to 1.1 MPa for the H-veneer after hydrothermal treatment. After pressing, the tensile strength rises to 18.1 MPa substantially. As for the Ni-plated H-veneer, it manifests the highest tensile strength of all the samples: 21.9 MPa. Notably, the tensile strength of the Ni-plated H-veneer is around 4.6 times that of carbon cloth (4.8 MPa) and 119 times that of graphene foam (0.184 MPa) [28]. Tensile tests of the veneer samples across the channel direction were also performed, and the results are plotted in Figure 4b. The tensile strength values of the P-veneer, H-veneer without pressing, and H-veneer after pressing across the channel direction are 30‒35% of that along the channel direction. This phenomenon is common for wood-based materials. The Ni-plated H-veneer also follows this trend—that is, the tensile strength across the channel direction declines to 4.4 MPa. Nevertheless, this value approaches that of carbon cloth, and is around 24 times that of graphene foam [28].

The electrical conductivity of the Ni-plated H-veneer was estimated by a two-probe method at room temperature, and the obtained *I*‒*V* curves are shown in Figure 5a. According to the curves, the electrical conductivity of the Ni-plated H-veneer is 1.05 × 10^3^ and 1.15 × 10^3^ S m^−1^ across and along the channel direction, respectively. For comparison, carbon cloth (W0S1002, CeTech Co., Ltd., Taiwan, China) was also tested under the same conditions, and its electrical conductivity was estimated to be 1.83× 10^3^ S m^−1^. Therefore, the Ni-plated H-veneer has a similar electrical conductivity to the widely used carbon cloth current collector. Furthermore, the electrical conductivity of the Ni-plated H-veneer is slightly higher than that of the graphene foam [29]. Moreover, the electrical conductivity of the Ni-plated H-veneer is significantly higher than that of previously reported wood-based conductors, such as PANi-modified wood veneer (1 × 10^−2^ S m^−1^) [16], PPy-coated wood veneer (1.92 × 10^2^ S m^−1^) [17], and a wood veneer‒PANi composite (9.23 × 10^−1^ S m^−1^) [18]. Four-probe measurements were also conducted for the Ni-plated H-veneer, revealing that the sheet resistances are 1.13 and 1.11 Ω sq^−1^ across and along the channel direction, respectively. Such values are approximately one-tenth that of conductive indium tin oxide (ITO) and Ag nanowires-decorated transparent wood veneer [30], being low enough for a current collector.

In addition, we used different electroless plating times to prepare the Ni-plated H-veneer. The corresponding electrical conductivity and Ni weight percentage values are plotted in Figure 5b. These values keep rising as the electroless plating time increases. If the Ni-plated H-veneer is used as the current collector for flexible EES devices, its weight should be as low as possible. As discussed above, when the electroless plating time is 6 h, a rather high electrical conductivity can be achieved, whereas a large amount of Ni is deposited (the weight percentage of Ni is 54.5%). Therefore, in order to balance the electrical conductivity and weight, we chose 6 h as the plating time in this study. Moreover, it should be mentioned that other types of wood can also be used as the precursor to construct Ni-plated veneers. For example, we tried using basswood and paulownia wood as the precursor. Although a large tensile strength and high electrical conductivity can also be realized for these products, the flexibility is much worse than when using balsa wood as the precursor. Therefore, we focus on a balsa wood-derived Ni-plated H-veneer in this work.

## 4. Conclusions

Thanks to its unique advantages, wood is now widely used as a raw material to construct functional materials. In this study, we developed a two-step process to transform natural balsa wood into flexible and conductive Ni-plated veneers. First, most of the lignin and hemicellulose in the wood precursor were removed through hydrothermal treatment. Then, Ni particles were uniformly grown on the lumen walls via electroless plating. Finally, the lumens were flattened after pressing. Along the channel direction, the obtained product possessed a much higher tensile strength than that of carbon cloth and graphene foam. In addition, our product also showed a high tensile strength across the channel direction. Furthermore, our product has a high electrical conductivity around 1.1 × 10^3^ S m^−1^, which is comparable to that of carbon cloth and graphene foam, and considerably higher than that of previously reported conductive wood-based veneers. Therefore, we have successfully constructed a new functional wood with great flexibility, large tensile strength, and high electrical conductivity using a scalable and cost-effective strategy. The obtained Ni-plated veneer can act as the substrate for the growth of various electrode materials and thus can be directly used as electrodes for flexible and wearable EES devices. Moreover, the methodology in this study can be extended to other types of wood with different deposited materials for a variety of applications.

## Figures and Tables

**Figure 1 materials-12-03198-f001:**
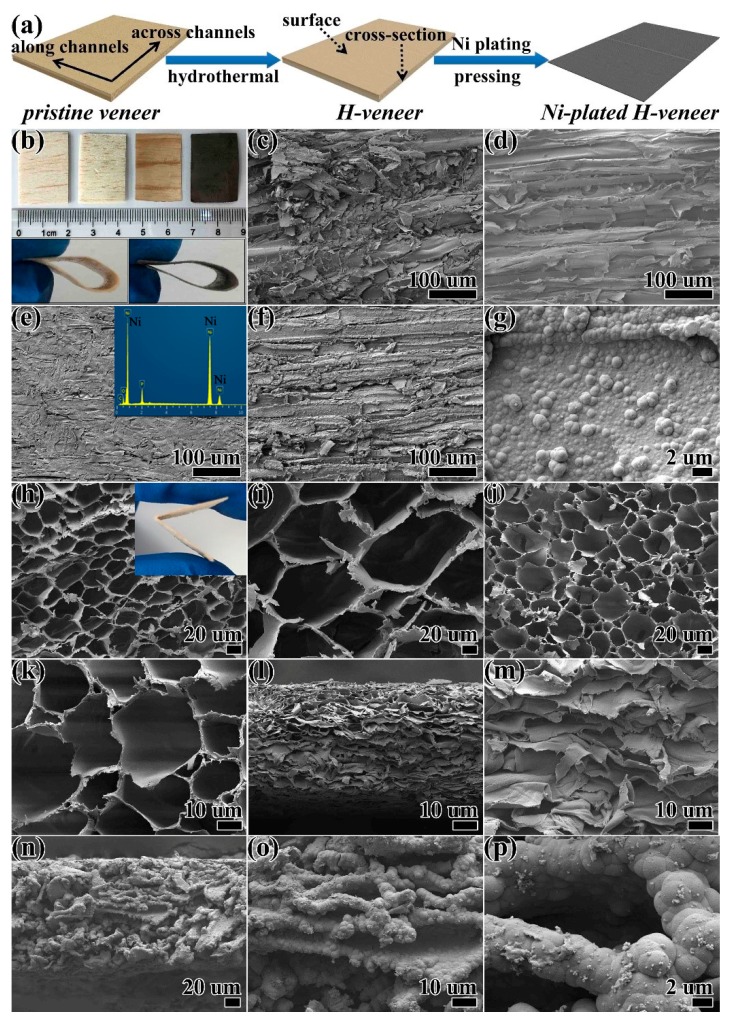
(**a**) Schematic illustration of the preparation of the Ni-plated H-veneer. (**b**) Photographic images of the four veneer samples put together (top), the H-veneer after pressing under bending (bottom left), and the Ni-plated H-veneer under bending (bottom right). Surface SEM images of (**c**) the P-veneer, (**d**) the H-veneer without pressing, (**e**) the H-veneer after pressing, and (**f**,**g**) the Ni-plated H-veneer. Cross-sectional SEM images of (**h**,**i**) the P-veneer, (**j**,**k**) the H-veneer without pressing, (**l**,**m**) the H-veneer after pressing, and (**n**–**p**) the Ni-plated H-veneer. The inset of (**e**) shows the EDS spectrum of the Ni-plated H-veneer. The inset of (**h**) shows a photograph of the P-veneer under bending.

**Figure 2 materials-12-03198-f002:**
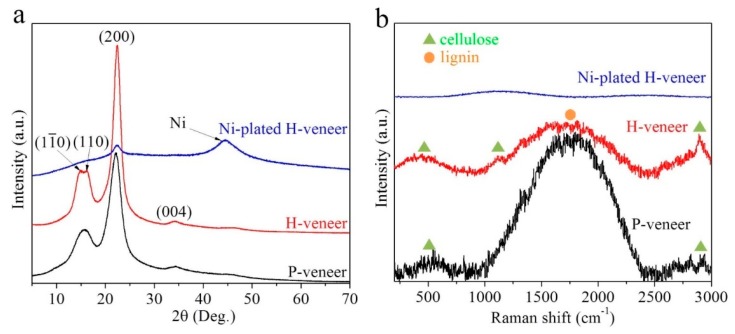
(**a**) XRD patterns and (**b**) Raman spectra of the P-veneer, H-veneer, and Ni-plated H-veneer.

**Figure 3 materials-12-03198-f003:**
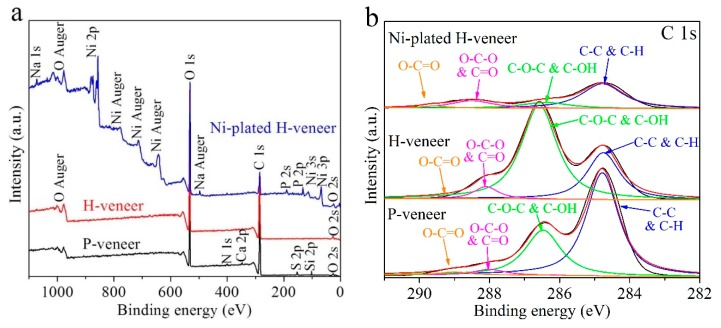
(**a**) Survey and (**b**) high-resolution C 1s XPS spectra of the P-veneer, H-veneer, and Ni-plated H-veneer.

**Figure 4 materials-12-03198-f004:**
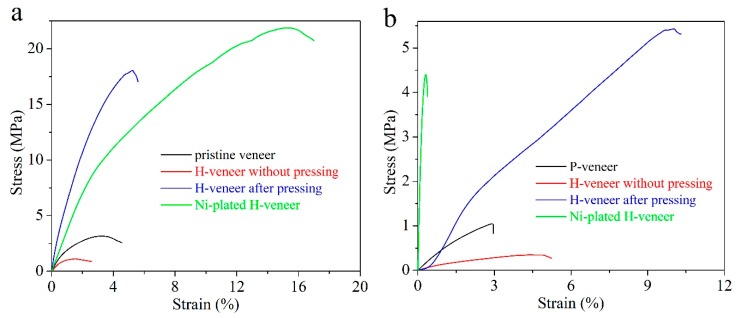
Tensile test results of the P-veneer, H-veneer without pressing, H-veneer after pressing, and Ni-plated H-veneer (**a**) along and (**b**) across the channel direction.

**Figure 5 materials-12-03198-f005:**
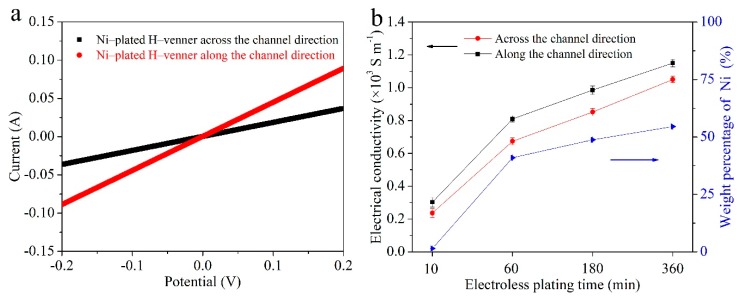
(**a**) *I*‒*V* curves of the Ni-plated H-veneer (6 h electroless plating) in two different directions. (**b**) The electrical conductivity of the Ni-plated H-veneer and the weight percentage of Ni in the Ni-plated H-veneer with respect to different electroless plating times.

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
