# Peer review of "Rendering Wood Veneers Flexible and Electrically Conductive through Delignification and Electroless Ni Plating"

_materials, 2019, doi:10.3390/ma12193198_

Round 1

Reviewer 1 Report

The article (Manuscript Number materials-590753) presents information about a two-step process to transform natural balsa wood into flexible and conductive Ni-plated veneers. In the first step, most of lignin and hemicellulose in the wood veneer are removed by hydrothermal treatment. Then, Ni particles are uniformly grown on the lumen walls via electroless plating. Finally, the lumens are flattened after pressing. As authors stated the obtained product possesses a much higher tensile strength than that of carbon cloth and graphene foam. The obtained Ni-plated veneer was flexible and bendable. Moreover, this product exhibits high electrical conductivity, which is comparable to that of carbon cloth and graphene foam, and significantly outperforms previously reported wood-based conductors. Thus, this product is particularly appealing for current collectors of flexible and wearable electronics.

On the whole, the manuscript is fairly well-written and logically arranged. The overall originality of the review concept used here is medium-high. Nevertheless, I would recommend publication of this review article in Materials on the condition a minor revision of the manuscript will be carried out and the following points will be taken into consideration.

Detailed comments:

The abstract needs to be well written with future prospects of the work and describe in short the concept of transformation of natural balsa wood into flexible and conductive Ni-plated veneers. More detailed advantage of the present field must be mentioned in the Introduction. The conclusion reflects an overall summary of the field with further extension and includes future prospective - I would suggest clarifying this section. The chapter appears to be a collection of data from research papers, however, authors self-opinion is of importance while drafting a chapter of this type.

After completing the above-mentioned corrections this work will be more readable. Therefore, it will be useful for the readers of the Materials.

Reviewer 2 Report

The manuscript is interesting with sound characterization studies of birch wood veneers. There are a couple of technical flaws in the manuscript. I would recommend a major revision of the manuscript with the following points:

1) What is the electromagnetic shielding effectiveness of the metallized wood veneers? The authors can show the influence on electromagnetic shielding effectiveness before and after electroless Ni plating.

2) The authors should calculate the nickel metal deposition amount on the wood veneers.

3) The authors should discuss about the relationship between electroless Ni metal deposition and surface resistivity on wood veneers.

4) If possible, EDS analyses of the plated wood veneers would be very useful to clearly show the nickel signals.

5) Have the authors tried to see the full potential window from -1 V to +1 V in the i-V plot of Figure 5? If not, the authors should explain why they picked the narrow potential window of -0.2 V to +0.2 V.

Reviewer 3 Report

The manuscript by Chen et al. describes a simple and cheap method for wood treatment to improve its mechanical and electrical properties for a variety of applications. The technological process is described in sufficient details and the authors also characterized the properties of the obtained wood product, called H-veneer. While the manuscript presents interesting results it fails to explain how authors arrived at the protocol they are using. How was the protocol optimized? And what happens if these technological steps are altered or skipped. The authors should describe the protocol optimization in details before the manuscript can be accepted. It is also missing the rationale on why this particular wood, balsa wood, was selected for this work. Can other types of wood be used and if so how they are better or worse?

The manuscript also requires some further editing to improve grammar and correct typos. Here are a few examples.

L24, "research", no plural form. L124 "are located", passive voice. L160 "H-veneer", typo. L176 what are the units?

Round 2

Reviewer 2 Report

The manuscript is extensively revised by the authors. I would recommend the minor revision of the following points before a decision can be made towards acceptance:

On page 6, lines 178 and 180, the authors state that a carbon cloth was tested in similar condition. They should mention the source of this carbon cloth here. Is it the same as described before? In Figure 3 b), please include C1s on the top right corner of the XPS spectrum. On page 6 line 186, please remove the character “口” and insert the correct letter/units.

Reviewer 3 Report

The authors address all my concerns and I have no further comments. The manuscript can be accepted for publication.
